# IoT-Enabled Few-Shot Image Generation for Power Scene Defect Detection Based on Self-Attention and Global–Local Fusion

**DOI:** 10.3390/s23146531

**Published:** 2023-07-19

**Authors:** Yi Chen, Yunfeng Yan, Xianbo Wang, Yi Zheng

**Affiliations:** 1College of Electrical Engineering, Zhejiang University, Hangzhou 310058, China; morningone@126.com; 2School of Mechanical Engineering, Zhejiang University, Hangzhou 310027, China; yvonnech@zju.edu.cn; 3Hainan Institute of Zhejiang University, Sanya 572025, China; xianbowang@zju.edu.cn

**Keywords:** few-shot image generation, power scenarios, self-attention encoder, multi-view feature fusion

## Abstract

Defect detection in power scenarios is a critical task that plays a significant role in ensuring the safety, reliability, and efficiency of power systems. The existing technology requires enhancement in its learning ability from large volumes of data to achieve ideal detection effect results. Power scene data involve privacy and security issues, and there is an imbalance in the number of samples across different defect categories, all of which will affect the performance of defect detection models. With the emergence of the Internet of Things (IoT), the integration of IoT with machine learning offers a new direction for defect detection in power equipment. Meanwhile, a generative adversarial network based on multi-view fusion and self-attention is proposed for few-shot image generation, named MVSA-GAN. The IoT devices capture real-time data from the power scene, which are then used to train the MVSA-GAN model, enabling it to generate realistic and diverse defect data. The designed self-attention encoder focuses on the relevant features of different parts of the image to capture the contextual information of the input image and improve the authenticity and coherence of the image. A multi-view feature fusion module is proposed to capture the complex structure and texture of the power scene through the selective fusion of global and local features, and improve the authenticity and diversity of generated images. Experiments show that the few-shot image generation method proposed in this paper can generate real and diverse defect data for power scene defects. The proposed method achieved FID and LPIPS scores of 67.87 and 0.179, surpassing SOTA methods, such as FIGR and DAWSON.

## 1. Introduction

The significance of defect detection in power equipment is crucial for the safety and stable development of both a country and society at large. With the emergence of the Internet of Things (IoT) [1], the integration of IoT with technologies such as artificial intelligence (AI) and deep learning (DL) is driving the advancement of machine detection methods.

The power industry embodies a highly intricate and hazardous environment, necessitating the imposition of stringent accuracy standards for machine inspection algorithms [2]. The defect detection method [3,4,5], which is based on deep learning, can independently extract multi-level and multi-angle features from the original data without artificial feature extraction. It can have higher detection accuracy and stronger generalization ability through feature learning. However, in order to improve the generalization ability and accuracy of the detection model, it relies on diverse and high-quality power scene defect data for training.

IoT devices [6,7], such as sensors and smart meters, can be deployed in power systems to gather real-time information on various parameters, such as voltage, current, temperature, and other relevant factors [8,9,10]. These IoT devices continuously collect data from the power scene, providing valuable insights into the operation and performance of the power equipment.

The present defect dataset for electrical equipment exhibits two primary limitations. First, a comprehensive dataset encompassing diverse defect categories on a large scale is currently inadequate. The existing defect dataset [11,12] suffers from an imbalanced distribution of images across different defect categories, and the total number of samples is limited. These shortcomings significantly compromise the effectiveness of the detection model. Second, defect image acquisition is difficult. Typically, image acquisition in the electric power domain involves issues related to data protection and security, thereby constraining the diversity of the scene variability. Such limitations impede the enhancement of the generalization capacity of the model, thereby hindering its efficacy. Based on the above challenges, this paper uses a few-shot image generation method to enrich the defect data of power equipment.

Few-shot image generation methods [13,14,15,16] are mainly divided into three categories: optimization-based methods, metric-based methods, and fusion-based methods. The optimization-based method [17,18] is similar to the migration model, where images of known categories are used to train the model, and images of unknown categories are used to fine-tune the model to achieve image generation. Inspired by matching networks, metric-based methods combine the matching process with VAEs [19,20]. Fusion-based methods fuse high-level and low-level features and then decode the features back to realistic images of the same class. Electrical scenes can be complex and dynamic, with multiple objects and lighting conditions that can change rapidly. This can make it difficult to capture a wide variety of defects and produce high-quality images consistent with the entire scene. Simultaneously generating defect images in power scenarios requires accurate detection and localization of defects. Compared with the former two, the fusion-based method has improved visual quality, better diversity, faster training speed, and better generalization ability. Therefore, this paper realizes the efficient generation of power equipment defect images based on the fusion-based small-sample image generation method.

In order to better realize the feature extraction of defects in complex power scenarios, a generative adversarial network based on **multi**-**view** fusion and **self**-**attention** is proposed (MVSA-GAN). Self-attention [21] can selectively fuse features from multiple generators according to the correlation with the input image, improving the diversity and authenticity of generated images. In addition, the complexity of the electrical scene and the richness of defect types should be reflected in the generated images at the same time. The fusion of global and local features, on the one hand, allows the model to generate images that are consistent with the overall scene. On the other hand, by fusing features of different spatial scales, the model is able to capture fine and coarse details of the input and improve the visual quality of the generated images.

Our contributions can be summarized as follows:A self-attention encoder (SAEncoder) is designed to help improve the quality of generated images via a more robust encoding of input images. By leveraging the self-attention mechanism, the encoder can effectively capture high-level features and contextual information of the input image, which can then be utilized to guide the image generation process.A global–local fusion module (GLFM) is designed to enhance the quality, diversity, and realism of generated images. By capturing both global and local features of input images, the GLFM can effectively guide the image generation process, enabling the creation of high-quality images with improved visual fidelity.Our study addresses the challenges of defect detection in electric power scenes by proposing a novel few-shot image generation network that leverages the self-attention mechanism and multi-view fusion, named MVSA-GAN. Specifically, our proposed network is designed to generate high-quality images with clear defect positions and diverse backgrounds, even when the available power defect scene data are limited.

The rest of the paper is arranged as follows: The current methods of generative adversarial networks (GANs) and few-shot image generation are described in Section 2. The proposed method for multi-exposure image fusion is introduced in Section 3. Additionally, the effectiveness of this method is verified through a computer simulation in Section 4. Finally, the conclusion is described in Section 5.

## 2. Related Works

### 2.1. Generative Adversarial Networks

GAN [22] is a branch of the generative model, which trains the model through adversarial learning, and uses mutual game learning of the generative model and the discriminative model to produce a fairly good output. Its training process is as follows: (1)$$\min_{G}\max_{D}V\left(D,G\right)=E_{\left[x∼real\right]}\log^{D\left(x\right)}+E_{\left[x∼fake\right]}\log^{\left(1-D\left(G\left(z\right)\right)\right)}$$
where *G* represents the unknown parameter set in the generator, *D* represents the unknown parameter set in the discriminator, *E* represents the expectation, and V(D,G) represents the cross-entropy function of the entire training process. Overall, for the function V(D,G), the parameter set of *G* is regarded as a symbolic constant first, and the parameter set *D* of the discriminator is regarded as a variable to find the maximum value of the function *V*. The maximum value is a symbolic constant algebraic expression containing *G*. Then, we find the minimum value of the function to obtain the optimal parameter value of the generator [23,24].

Due to the advantages of GANs in fitting data distributions, it is effective in image generation, image editing, and image–image translation [25,26]. The reason is mainly due to the unlimited supply of training images, which requires a large amount of training data as support. However, in the case of limited data, the discriminator is prone to overfitting, which makes it difficult for the model to converge [27,28]. Recently, some researchers [29,30] have proposed some advanced data augmentation strategies for training GANs with limited data, but these methods are mainly designed for unconditional generation. To solve the disadvantages of generative adversarial network training on limited datasets, this paper attempts to solve this problem with the few-shot learning paradigm. The GAN network generates different images for a new category by limiting the data category; we hope that GAN can generate large numbers of real and diverse images with only a few images in this category.

### 2.2. Few-Shot Image Generation

Few-shot image generation is a subfield of computer vision that aims to generate high-quality images from a small number of training examples. The goal is to develop algorithms that can generate realistic images of objects, scenes, and other visual content with a limited amount of data. Optimization-based methods [31,32] involve optimizing a model to generate high-quality images by minimizing a loss function that measures the difference between the generated and real images. These methods typically involve training a generative model, such as a generative adversarial network (GAN), to generate images from a small number of input examples. FIGR [18] and DAWSON [33,34] combine adversarial learning with meta-learning methods (i.e., reptile [35] and MAML [36]), but the quality of the generated images is low and not suitable for practical applications. Metric-based methods [37,38] involve learning a distance metric that can measure the similarity between images in a high-dimensional feature space. The generator can then generate new images by finding the closest image in the feature space to the input examples. Some notable metric-based methods include Siamese networks [39,40], which learn a distance metric between images in a feature space, and prototypical networks [41], which use prototypes to represent classes in a feature space. Fusion-based methods [16,42,43] involve combining global and local features of the input images to generate high-quality images.

GMN [44] and MatchingGAN [45] use VAE and GAN to generalize the matching network from a small sample classification task (few-shot classification) to a small sample generation task. F2GAN [46] improves MatchingGAN by adding a non-local attention fusion module to fuse and fill features at different levels to generate images. LoFGAN fuses deep features at a finer level by selecting, matching, and replacing local representations, and uses a local-based reconstruction loss to reduce aliasing artifacts. However, the above methods only focus on local features, and the power scene is complex and dynamically changing. A single change in the local features is difficult to apply to the actual power equipment defect detection task.

## 3. MVSA-GAN

### 3.1. Overall Framework

The overall framework of MVSA-GAN is shown in Figure 1, which includes four modules: IoT data collection, SAEncoder, GLFM, decode, and discriminator. First, sensors and devices in the IoT are used to collect various types of power scenario data, such as current, voltage, temperature, etc. These data can be combined with the subsequent MVSA-GAN to help generate synthetic images with features of electric scenes. Second, a small number of training images are input into the encoder, and operations, such as GLFM and encoder, are used to generate similar images. Then the input image and the generated image pass through a discriminator to improve the authenticity of the generated image. Class loss $L_{cls}$ and the adversarial loss $L_{adv}$ are used to update the parameters in the discriminator. Finally, the input image, the generated image, and the output features of GLFM undergo a global–local loss $L_{glf}$ calculation to improve the robustness and authenticity of the generative model.

The SAEncoder can extract more feature information from limited samples. By using the Transformer structure, it can capture contextual features, making the features extracted by the encoder more diverse and effective. This allows for better modeling and learning of details and critical parts of electrical defect scenarios, resulting in more realistic and accurate images. Secondly, GLFM can fuse features from different scales, so that the generated images have more comprehensive and rich information. In the power defect scenario, defects are usually distributed in different locations and scales, so it is necessary to fully utilize and fuse features of different scales. Through the multi-view fusion module, we can better combine features of different scales, so that the generated images have more accurate and comprehensive defect information.

### 3.2. SAEncoder

A context encoder based on the self-attention mechanism is proposed, named SAEncoder, which uses the ability of the Transformer structure to capture context features to enrich the features extracted by the encoder. SAEncoder can improve the authenticity of subsequently generated images by learning image details. It consists of five stages, and each stage consists of a context-based Transformer (CoT) module [47]. For the self-attention mechanism of the Transformer, each input unit can calculate a weight vector based on its relative position with other input units, which can be used to weigh the feature representation of the input unit. Therefore, each unit can take into account the contextual information of the entire input sequence, not just the part before or after it. This allows the encoder to extract more information from limited samples and generate more realistic images. In addition, since each input unit can take into account the contextual information of the entire input sequence, it can better capture the long-term dependencies, and better capture the correlation and continuity between images, making the generated images more natural.

CoT is a Transformer module, and we use the Transformer structure to improve the ability of the encoder (SAEncoder) to capture contextual features. The self-attention mechanism plays a key role in CoT. The self-attentiveness mechanism works as follows: (2)$$Att\left(Q,K,V\right)=\mathrm{softmax}(\frac{QK^{T}}{\sqrt{d_{k}}})V$$
where *Q* stands for query, which represents the query features of the *i*-th patch learned by the neural network, *K* stands for key, which represents the queried features of the *j*-th patch learned by the neural network, and *V* stands for value, which represents the semantic features learned by the neural network from the *j*-th patch. The self-attentive mechanism adaptively learns and matches the features learned by the neural network by such words.

The structure of CoT is shown in Figure 2. First, a 3 × 3 convolution performs contextual encoding on the input *K* to obtain a static contextual representation of the input. Further, the concat operation is implemented on the encoded *K* and the input *Q*, and the dynamic multi-head attention matrix is learned through two consecutive 1 × 1 convolutions. The learned attention matrix is multiplied by the input values to achieve a dynamic contextual representation of the input. Finally, we take the results of the static and dynamic contextual representations as output. The input *K* in the figure contains a large amount of contextual information, through which the learning of the dynamic attention matrix can be optimized, and the feature representation ability of the network can be further enhanced.

The hierarchical structure of the SAEncoder is shown in Table 1. In the first stage, the dimension of the input is first increased and the scale of the image is kept unchanged, and then the scale of the input image is reduced, step by step, through four stages, reducing the calculation pressure, and extracting high-dimensional features. In the last four stages, this paper constructs a block structure, which contains a 1 × 1 convolution, a CoT module, and a 3 × 3 convolution in the block. The former two keep the dimension of the feature unchanged to extract deep features in the feature map, and the latter uses a 3 × 3 convolution to increase the dimension of the feature map. Such a structure can improve the feature extraction ability of the encoder, and further enable the generator to generate images with real image details.

### 3.3. Global and Local Fusion Module

The existing local fusion module (LFM) [16] randomly chooses one of the encoded features as the base and the rest of the features as references, and fuses them by local selection, local matching, and local replacement. The LFM module takes the output feature map of the encoder as input, randomly selects one of the feature maps as the base feature $f_{base}$, and uses the other feature maps as the reference features $F_{ref}$. For example, when the LFM module performs a five-shot image generation task, the LFM will use the remaining four feature maps as $F_{ref}$. The local fusion module will take the select $f_{base}$ as the basis and the rest $F_{ref}$ as a bank of local features to produce a fused feature. The entire fusion process of LFM can be divided into three steps, including local selection, local matching, and local replacement.

In the local selection stage, *n* local representations in $f_{base}$ should be replaced randomly. After the local selection ends, *n* c-dimensional local representations $\Phi_{base}$ from the base feature $f_{base}$ will be obtained. In the local matching stage, LFM will look for matching local representations in $F_{ref}$, which can replace $\Phi_{base}$ semantically. We calculate the similarity matrix *M* of $\Phi_{base}$ and $f_{ref}$ by Equation (Equation 3). The feature map $\Phi_{ref}$ that best matches $\Phi_{base}$ can be obtained through the similarity matrix *M*.
(3)$$M^{\left(i,j\right)}=g\left(\phi_{base}^{\left(i\right)},f_{ref}^{\left(j\right)}\right)$$
where i∈{1,⋯,n}, j∈{1,⋯,h×w}, and *g* is a similarity metric, $f_{ref}\in F_{ref}$.

In the local replacement stage, the $\Phi_{ref}$ and $\Phi_{base}$ obtained in the local matching stage are weighted and summed to obtain the final fused local representation. Finally, these fused local representations are replaced on $f_{base}$, and the fused feature map $F_{fuse}$ is obtained as the output of LFM.

For each c-dimensional local representation in $\phi_{base}$, we now have k−1 candidate local representations. For example, $\phi_{ref}^{\left(1\right)}\in R^{(k-1)\times c}$ contains the most similar local representations with the first local representation $\phi_{base}^{\left(1\right)}\in R^{c}$, which we can find in every $f_{ref}$ (see the dotted lines in the LFM module in Figure 3). We fuse all of these local representations together and replace them with the corresponding positions in $f_{base}$. We use a random coefficient vector $a=[a_{1},\cdots ,a_{k}]$ to fuse the features for all the positions selected,
(4)$$\phi_{fuse}^{\left(t\right)}=a_{base}\cdot \phi_{base}^{\left(t\right)}+\sum_{i=1,\ldots ,k,i\neq base}a_{i}\cdot \phi_{ref}^{\left(i\right)}\left(t\right)$$
where $\Sigma_{i=1}^{k}a_{i}=1$, $a_{i}\geq 0$, and t=1,⋯,n. We retain original local representation with a ratio, $a_{base}$. Then, we replace all the *n* fused local representations $\phi_{fuse}$ with the corresponding positions in $f_{base}$. This produces a fused feature map *F* as the output of the LFM module.

LFM can well-fuse the local features of each feature map, generate a representative fusion feature map, and enhance the diversity of the generated network. However, LFM only considers the local features of the feature map, the fused feature map lacks global representation, and the semantic information of the generated fused feature map still does not change significantly, which will cause the model to be prone to overfitting.

In response to this problem, a global and local fusion module (GLFM) is proposed in this paper, which incorporates a global representation into the fusion of local representations. It greatly integrates the global semantic information of $F_{ref}$, making the fused feature map more representative, and further training a more robust small-sample generation model. As shown in Figure 3, GLFM adds a Transformer block to extract the global representation of the input feature map on the basis of the LFM module. This global representation represents the semantic information of $f_{ref}$. We set a random coefficient vector $b=[b_{1},\cdots ,b_{k}]$ to deeply fuse the generated global representation $G=[G_{1},\cdots ,G_{k}]$ with $f_{local}$, where *k* stands for *k*-shot image generation. Also, we set a threshold parameter τ to avoid the impact of unimportant global representation on the fusion of local representations. The calculation for $f_{glf}$ is as follows: (5)$$f_{glf}=\left(1-\tau \right)\cdot f_{local}+\sum_{i}^{k}b_{i}\cdot G_{i}$$
where τ and *b* are variable parameters, and the parameters are adaptively adjusted according to the feature map input by GLFM.

## 4. Experiments

### 4.1. Datasets

In this paper, a total of 59,169 images of defects in various scenes of substations were collected. The defect categories include blurred dials, damaged dials, cracked insulators, and metal corrosion. The collected defect images are divided into 12 categories, and the specific categories and image data information are shown in Table 2.

To demonstrate that the MVSA-GAN can generate realistic and rich images in the case of limited samples, this paper randomly selected 400 images from each category, and a total of 4800 images constitute the training dataset. At the same time, in the training process, 10 image categories were used as visible images during training, and the other 2 categories were used as invisible images.

### 4.2. Experiment Details

The experimental environment configuration for the proposed method in this paper is based on Ubuntu16.04.1 (Linux 4.15.0-142-generic). We used an NVIDIA GPU (GeForce RTX 3090) with 24 GB of video memory and 256 GB of system memory. The batch size for training was set to 16, with a total of 100,000 iterations.

An IoT-based platform was established to evaluate the MVSA-GAN applied to defect detection in power scenarios, as shown in Figure 4. The platform integrates various sensor devices to monitor the operating status of the power equipment and systems in real time. By collecting data from different sensors, the platform is able to generate image samples related to power scenes. Real-time sensor data collected through the IoT platform provide key information for small-sample image generation. These data can be combined with existing electrical equipment defect image samples to generate additional defect image samples through a generative algorithm. Therefore, a large number of power equipment defect images can be simulated and generated without actual defects, facilitating the training and testing of defect detection algorithms.

### 4.3. Evaluation Metrics

In the experimental part, four image quality evaluation metrics are used to quantitatively evaluate the proposed method, as follows:

The Frechet inception distance (FID) score [48] is used to calculate the distance between the real data $S_{real}$ and the generated data $S_{gen}$. The similarity between the two sets of images is measured by the statistical similarity of the visual features of the original image. In general, the mean and covariance matrices are used to calculate the distance between two distributions, as follows: (6)$$FID\left(x,g\right)={∥\mu_{x}-\mu_{g}∥}_{2}^{2}+Tr\left(\Sigma_{x}+\Sigma_{g}-2{\left(\Sigma_{x}\Sigma_{g}\right)}^{\frac{1}{2}}\right)$$

Learned perceptual image patch similarity (LPIPS) [49] is the similarity of learning perceptual image patches, which can measure the difference between two images. The lower the value, the more similar the two images are. The calculation formula is as follows: (7)$$d\left(x,x_{0}\right)=\sum_{l}\frac{1}{H_{l}W_{l}}\sum_{h,w}{∥w_{l}⊙\left({\hat{y}}_{hw}^{l}-{\hat{y}}_{0hw}^{l}\right)∥}_{2}^{2}$$
where *d* is the distance between $x_{0}$ and *x*. Extract the feature stack from the *L* layer and perform unit normalization in the channel dimension (unit-normalize). Use the vector $W_{L}$ to scale the number of activated channels and calculate the $L_{2}$ distance. Finally, we average over the space and sum over the channels.

The inception score (IS) [50] is used to evaluate the difference between two distributions, and the calculation formula is as follows: (8)$$IS\left(G\right)=exp\left({E_{x}}_{∼pg}D_{KL}\left(p\left(y|x\right)\left|\right|p\left(y\right)\right)\right)$$
where $E_{x∼pg}$ means to traverse all generated samples and find their average. $D_{KL}$ represents the KL-divergence, and $D_{KL}\left(p\left(y|x\right)\right)$ is used to measure the degree of approximation between the distribution *P* and *Q*. p(y|x) represents the probability distribution of all categories for the picture *x*. p(y) is marginal probability.

The peak signal-to-noise ratio (PSNR) [51] is used to evaluate the noise contained in the image; the calculation formula is as follows: (9)$$PSNR=10\log_{10}\left(\frac{MAX_{I}^{2}}{MSE}\right)$$
(10)$$MSE=\frac{1}{mn}\sum_{i=0}^{m-1}\sum_{j=0}^{n-1}{\left[I(i,j)-K(i,j)\right]}^{2}$$

The size of the original image *I* is mn, and the image *K* is generated after adding noise to it. $MAX_{I}$ represents the maximum pixel value of the image.

The F${}_{1}$ score [52] is introduced as an indicator for evaluating downstream tasks, and its calculation formula is as follows: (11)$$F_{1}=2\cdot \frac{precision\cdot recall}{precision+recal}$$

### 4.4. Evaluation Results

We compare MVSA-GAN with existing GAN-based methods, and the visualization results are shown in Figure 5. This method first judges the defect category of the input image to learn defect features, and then combines global and local features to generate corresponding virtual images. It can be seen from Figure 5 that the generated images are real and diverse, and have good processing effects on defect images in different scenarios. Furthermore, the model shows strong robustness in preserving the main features of original defects while incorporating features learned from images of other defect categories. The generated images are faithful to the original texture clarity and fidelity of the input images. By combining global and local features, generative models can capture complex and diverse textures, resulting in high-quality images that closely resemble the input data. The method’s ability to learn and incorporate features from images of other defect categories enhances its robustness and adaptability to various real-world scenarios.

Table 3 presents the quantitative comparative analysis between the MVSA-GAN and state-of-the-art (SOTA) methods, including FIGR [18], MatchingGAN [45], DAWSON [34], LoFGAN [16], GMN [44], and DAGAN [53]. Our method outperforms other methods in four key metrics. The proposed method achieves 67.87 and 0.179 on the FID and LPIPS, and 164.23 and 23.41 on the IS and PSNR. The experimental results show that the proposed method can capture complex textures and generate high-quality images that are very similar to the input data. In terms of IS and PSNR, it is 11.53 and 4.1 higher than the next highest method, respectively. It can be seen that the proposed method is robust and adaptable to complex power scenarios.

### 4.5. Ablation Experiment

In this section, we evaluate the effectiveness of the proposed GLFM and SAEncoder in the generation of the electrical scene defect images through ablation experiments, as shown in Table 4. Through the quality evaluation of the image generated by a single module and the joint action of the two, it is obvious that the authenticity of the image generated by the proposed method can be seen. In particular, after adding GLFM, the three indicators used for measuring image quality are significantly optimized. Under the joint action of GLFM and SAEncoder, the FID, IS, and PNSR of the final image generated by the model are 67.87, 164.23, and 23.41, respectively. It can be seen from the changes in the data that the proposed method can well adapt to complex power scenes, demonstrating its ability to capture and learn subtle defect features in the scene.

Figure 6 specifically shows the feature learning ability of the SAEncoder for the details in complex power defect scenes. For the encoder without the self-attention mechanism, it cannot capture comprehensive contextual information due to the limitation of the receptive field. This results in an incoherent generated image with blurred, distorted details. The SAEncoder establishes long-distance connections to global features through a self-attention mechanism, making image generation more natural.

Figure 7 provides a visualization of the effectiveness of GLFM in generating models. From Figure 7, the inclusion of GLFM enhances the generated image by making it clearer and more detailed. In the power scene defect image generation task, since there are many details and texture information in the image, the fusion of global features and local features is particularly important for generating high-quality images. Global features and local features have different advantages in extracting image information. Global features can capture the global structure and background information of the entire image, while local features can capture the detailed information of specific regions in the image. GLFM can lead to the interaction and integration of local features and global features, to better capture the structure and detailed information of the image and improve the realism and clarity of the generated image.

### 4.6. Downstream Tasks

To demonstrate that the MVSA-GAN has an enhanced effect on downstream tasks, in this paper, ResNet50 [54] and VGG16 [55] are used as classification networks, and training and evaluation are performed on datasets with and without synthetic data. The evaluation results are shown in Table 5. “Real” represents a small number of real samples, and “Sample” represents the data after image generation using the proposed method. The results show that the images generated by the generative model have a large improvement in the accuracy of the classification task. This is because the generative model can generate a large amount of virtual data, which enriches the sample diversity of the training dataset, helps the model to better learn the characteristics of the data, and improves the generalization ability of the model.

Since the power scene defect dataset is usually small in size, using real datasets for training may lead to overfitting of the model, which cannot effectively capture the characteristics of the data. The method proposed in this paper can increase the diversity of data by generating virtual data, which can help improve the generalization ability of the model and the accuracy of defect detection.

## 5. Conclusions

To address the scarcity of datasets for defect detection in electric power scenes, a generative adversarial network based on self-attention multi-view fusion (MVSA-GAN) is proposed for few-shot image generation. First, the power scene is complex and dynamic, and complex background features will cause artifacts in the generated image. SAEncoder is proposed to face the above challenge. By focusing on the relevant features of each part to capture contextual information, the generated image is more coherent and global, and effectively alleviates the artifacts and confusion generated by complex backgrounds. Secondly, to overcome the problem of existing small sample generation networks in generating images with both authenticity and diversity, GLFM is proposed. This module is used to selectively fuse global and local features, not only capturing background features but also extracting structure and texture details of defect parts, so as to achieve the authenticity and diversity of generated images. The proposed method provides high-quality data for electrical scene defect detection, achieving 67.87 and 0.179 on FID and LPIPS scores.

## Figures and Tables

**Figure 1 sensors-23-06531-f001:**
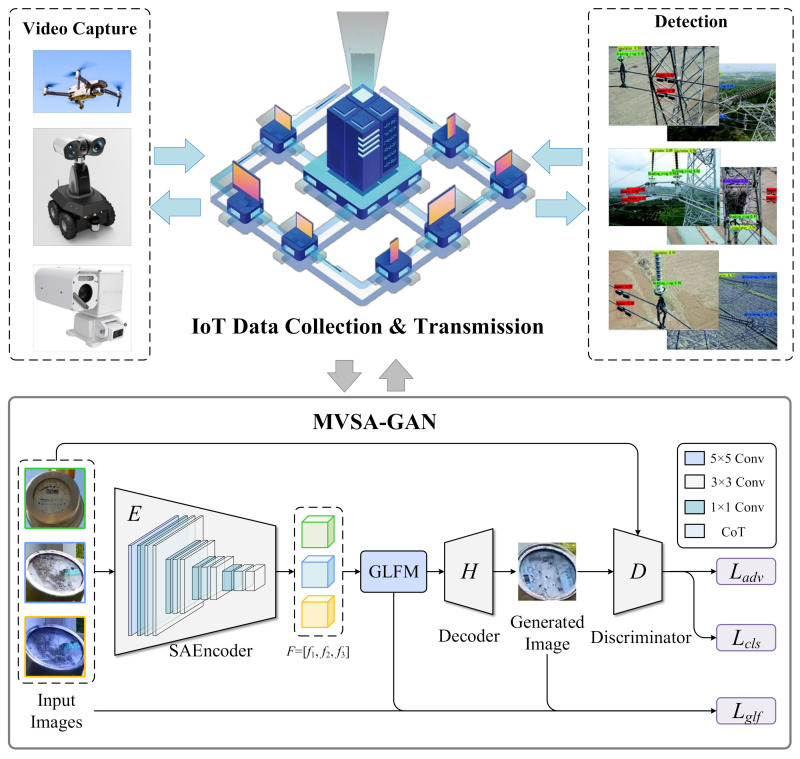
The overall structure of the MVSA-GAN.

**Figure 2 sensors-23-06531-f002:**
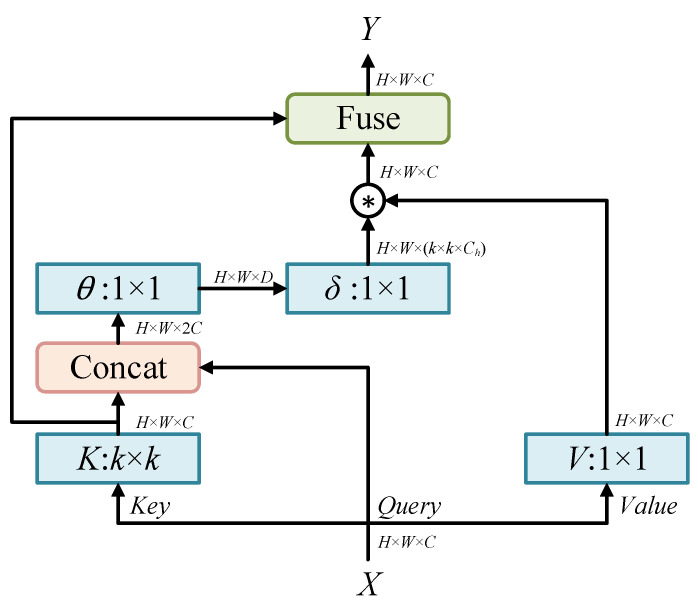
Context-based Transformer module structure.

**Figure 3 sensors-23-06531-f003:**
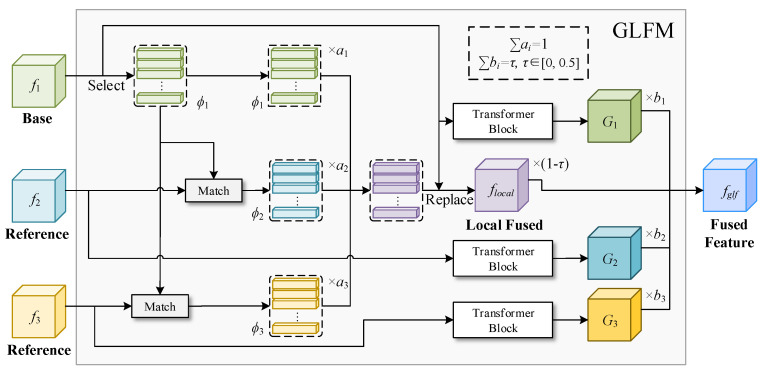
The overall structure of GLFM.

**Figure 4 sensors-23-06531-f004:**
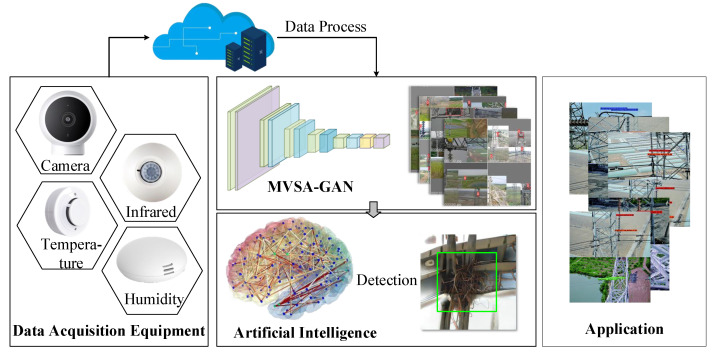
The IoT-based test platform for the evaluation of the MVSA-GAN.

**Figure 5 sensors-23-06531-f005:**
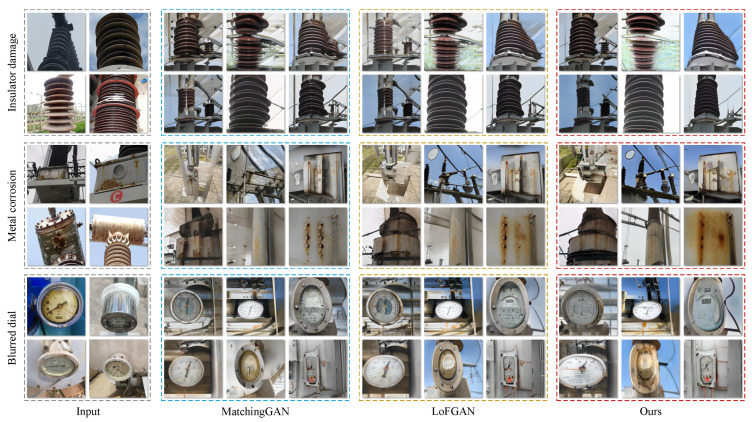
Generation samples of defect data in power scenarios by different methods.

**Figure 6 sensors-23-06531-f006:**
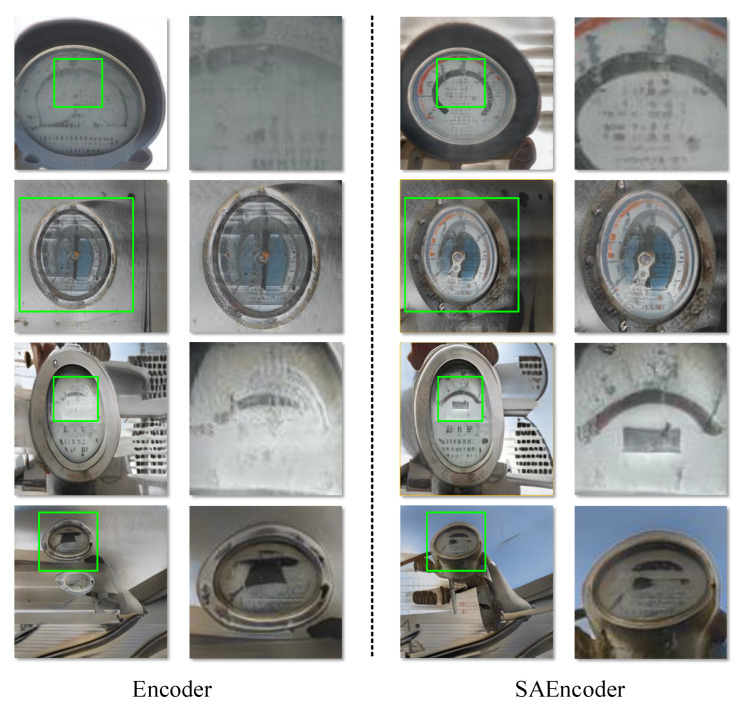
On the left, we can see the synthesis result obtained using the baseline encoder. On the right side, we can observe the synthesis result obtained using SAEncoder.

**Figure 7 sensors-23-06531-f007:**
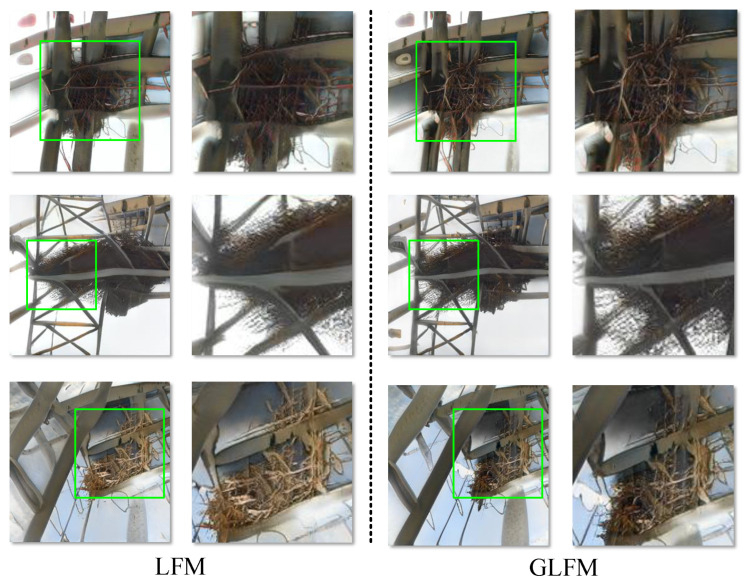
The left image shows the network synthesis result without GLFM, and the right image shows the network synthesis result using GLFM.

**Table 1 sensors-23-06531-t001:** Hierarchy of the SAEncoder.

	Encoder ^1^	Output Size
Stage 1	5×5, 32	128×128
Stage 2	$\left(\begin{array}{c} 1\times 1,32\\ CoT,32\\ 3\times 3,64 \end{array}\right)$	64×64
Stage 3	$\left(\begin{array}{c} 1\times 1,64\\ CoT,64\\ 3\times 3,128 \end{array}\right)$	32×32
Stage 4	$\left(\begin{array}{c} 1\times 1,128\\ CoT,128\\ 3\times 3,128 \end{array}\right)$	16×16
Stage 5	$\left(\begin{array}{c} 1\times 1,128\\ CoT,128\\ 3\times 3,128 \end{array}\right)$	8×8

^1^ This column presents information about the layers contained in the stage, and the cell provides the convolution kernel size and the number of channel layers of the convolution.

**Table 2 sensors-23-06531-t002:** Power scene defect data category information.

Category ID	Category Name	Number
1	Blurred Dial	1756
2	Broken Dial	533
3	Insulator Rupture	953
4	Insulator Crack	678
5	Oily Dirt on the Surface of Oil Leaking Parts	5922
6	Oil on the Ground	1494
7	Damaged Respirator Silicone Cartridge	527
8	Discoloration Silicone Respirator	3225
9	The Box Door is Closed Abnormally	1405
10	Suspended Solids	2006
11	The Bird’s Nest	1316
12	Damaged Cover	556

**Table 3 sensors-23-06531-t003:** Comparative experiment.

Method	The Smaller the Better		The Larger the Better	
	FID	LPIPS	IS	PSNR
FIGR	83.09	0.320	137.42	12.84
MatchingGAN	73.85	0.241	148.35	19.31
DAWSON	82.49	0.337	140.14	14.32
LoFGAN	86.98	0.362	133.95	12.47
GMN	79.14	0.292	141.33	14.85
DAGAN	75.75	0.265	152.70	18.46
Ours	**67.87**	**0.179**	**164.23**	**23.41**

**Table 4 sensors-23-06531-t004:** Ablation experiment.

	+GLFM	+SAEncoder	FID	IS	PSNR
Net.1			86.98	133.95	12.47
Net.2	✓		73.50	154.76	18.58
Net.3		✓	72.28	150.35	17.16
Net.4	✓	✓	**67.87**	**164.23**	**23.41**

**Table 5 sensors-23-06531-t005:** Optimization of model performance for downstream tasks with generated data.

		F${}_{1}$ Score			
Network	Dataset	Meter	Metal	Broken	Total
		Breakage	Corrosion	Insulator	
ResNet50	Real	89.0%	79.7%	90.4%	87.1%
ResNet50	Real+Sample	92.8%	84.4%	94.2%	90.8%
VGG16	Real	86.3%	78.3%	89.5%	85.6%
VGG16	Real+Sample	88.1%	85.0%	94.1%	89.4%

## Data Availability

The data generated during this study are currently private due to our team is about to have a new breakthrough in this study. The data will be made public soon.

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
