# Peer review of "IoT-Enabled Few-Shot Image Generation for Power Scene Defect Detection Based on Self-Attention and Global–Local Fusion"

_sensors, 2023, doi:10.3390/s23146531_

Round 1

Reviewer 1 Report

This paper has a designed self-attention encoder that focuses on the relevant features of different parts of the image to capture the contextual information of the input image and improve the authenticity and coherence of the image.

In my opinion it is really good paper

The level is okay

Author Response

Thank you for your recognition of the contributions made to this paper. We will continue to explore further valuable research based on our current results. 

Reviewer 2 Report

The paper is easy to read, well and clearly written, contains new results. The manuscript satisfies the rules of the journal. I guess it can be published.

Author Response

(The authors gave the same response as above.)

Reviewer 3 Report

1. The author’s manuscript said “Meanwhile, a generative adversarial network based on 7Multi-Scale fusion and Self-Attention is proposed for few-shot image generation, named MS2ANet.”.

Why is used by the MS2AN-Net in the reference[1]?

[1] Shahi, Kasra Rafiezadeh, et al., "MS2A-Net: Multiscale Spectral–Spatial Association Network for Hyperspectral Image Clustering," IEEE Journal of Selected Topics in Applied Earth Observations and Remote Sensing, vol. 15, pp. 6518-6530, 2022.

2. Please show the equation of the MS2ANet. 

3. The author’s manuscript said “A multi-scale feature fusion module is proposed to capture the complex structure and texture of the power scene through the selective fusion of global and local features, and improve the authenticity and diversity of generated images.”

Please show the equation of the multi-scale feature fusion module.

4. Please show the names of MS2ANett, FID, LPIPS, SOTA, FIGR, and DAWSON.

5. The author’s manuscript said “• A self-attention encoder (SAEncoder) is designed to help improve the quality of 75generated images by more robust encoding of input images.”

Please show the equation of the robust encoding of input images.

6. The author’s manuscript said “• A global-local fusion module (GLFM) is designed to enhance the quality, diversity and realism of generated images.”

Please show the equation of the GLFM.

7. The author’s manuscript said “• A global-local fusion module (GLFM) is designed to enhance the quality, diversity and realism of generated images.”

Please show the equation of the quality, diversity, and realism of generated images

8. The author’s manuscript said “We make the GAN network generate different images for a new category by limiting the data category`, and further we hope that GAN can generate a large number of real and diverse images with only a few images of this category.”

GAIN is used by the reference [18, 23, 24].

It is not good to publish the Sensors.

9. Please show the equation of the GAIN.

10. The author’s manuscript said “It consists of 5 stages, and each 174stage consists of a context-based Transformer module (CoT) [37].”

It is not good enough to publish the Sensors.

11. The author’s manuscript said “The feature map Φre f that best matches Φbase can be obtained 218through the similarity matrix M.”

It is the same as the reference [14].

12. The author’s manuscript said “The feature map Φre f that best matches Φbase can be obtained 218through the similarity matrix M.”

Please show example of the M(Φbase, Φre f).

13. Please show the difference of the Figure 3 and Figure 1 in reference [14].

14. The author manuscript said “where τ and b are variable parameters, and the size of the parameters is adaptively adjusted according to the feature map input by GLFM.”

Please show the equation of the size of the parameters.

15. The equation from the author's manuscript is too short.

Author Response

Thank you very much for your valuable feedback. We deeply regret the issue you raised regarding the similarity between our paper's title and another publication. We apologize for any confusion caused and would like to address this matter as follows:
 We will thoroughly examine the other paper with a similar title and ensure that our paper is distinct in terms of content and structure.
 The multi-scale fusion proposed in this paper represents the two-stage fusion of local features and global features. We will thoroughly examine the other paper with a similar title and ensure that our paper is distinct in terms of content and structure. In the revised manuscript, we have changed it to “a generative adversarial network based on Multi-View fusion and Self-Attention is proposed for few-shot image generation, named MVSA-GAN”

Reviewer 4 Report

The manuscript proposes a novel few-shot image generation neural network to produce high-quality images of power scene defect even in case of diverse backgrounds and limited numbers of data samples available. The quality of the research is high and it will without doubt be interesting for the readers of the journal. Furthermore, the manuscript has a good list of references and quality of the figures is good. Thus, I would recommend to accept the manuscript after correcting a few language issues.

Is an implementation of the neural network model proposed publicly available, e.g. on Github? If not, are you planning to make it available?

Is "on" missing in the title? Should it be: "Power Scene Defect Detection based on Self-Attention and Global-Local Fusion"?

Page 6, lines 217 - 218: "Calculate the similarity matrix M 217 of Φbase and fre f by formula", what formula?

Language in the manuscript is ok in general, but some phrases just sound funny, e.g.:

"SAEncoder ... use the Transformer structure to capture contextual features, making the features extracted by the encoder more contextual..." :)

Please double-check the language and correct if needed.

Author Response

We would like to take this opportunity to thank you for all your time involved and this great
opportunity for us to improve the manuscript. First of all, I would like to express my sincere regret to you and the deep learning community that we are not going to make the neural network model proposed public for the time being. The main reason involved is that we are still in the process of updating and iterating our algorithm, and we hope that our results will be even better and better. Again, we apologize for our plans. However, it is worth looking forward to the time when we plan to iterate this result to a certain extent, we will make it public, and hope that the majority of deep learning enthusiasts can join us together for the development of few-shot image generation network together.

Round 2

Reviewer 3 Report

The author's manuscript is good enough to publish the "Sensors".

no